# RNA-Mediated Regulation of Meiosis in Budding Yeast

**DOI:** 10.3390/ncrna8060077

**Published:** 2022-11-15

**Authors:** Vidya Vardhini Pondugala, Krishnaveni Mishra

**Affiliations:** Department of Biochemistry, School of Life Sciences, University of Hyderabad, Hyderabad 500046, India

**Keywords:** meiosis, cell fate, RNA processing, non-coding RNA, *Saccharomyces cerevisiae*

## Abstract

Cells change their physiological state in response to environmental cues. In the absence of nutrients, unicellular fungi such as budding yeast exit mitotic proliferation and enter the meiotic cycle, leading to the production of haploid cells that are encased within spore walls. These cell state transitions are orchestrated in a developmentally coordinated manner. Execution of the meiotic cell cycle program in budding yeast, *Saccharomyces cerevisiae*, is regulated by the key transcription factor, Ime1. Recent developments have uncovered the role of non-coding RNA in the regulation of Ime1 and meiosis. In this review, we summarize the role of ncRNA-mediated and RNA homeostasis-based processes in the regulation of meiosis in *Saccharomyces cerevisiae*.

## 1. Introduction

Meiosis is a complex biological process in sexually reproducing organisms that results in the formation of haploid gametes from a diploid progenitor cell. It is a reductional division achieved by one round of DNA replication in a diploid, followed by two consecutive cell divisions, namely meiosis I and meiosis II. The segregation of chromatin into a binucleate and then a tetranucleate state forming tetrads is accompanied by several structural, biochemical and metabolic changes [1]. In budding yeast, entry into meiosis has two pre-requisites: a favorable nutritional state and the cell type. *Saccharomyces cerevisiae* undergoes meiosis and forms stress-resistant spores when grown in a non-fermentable carbon source in the absence of a nitrogen source as an adaptation to harsh environmental conditions. This type of response to nutrient deprivation is observed only in heterozygous *MAT*a*/MATα* diploid cells and neither in haploid cells nor in *MAT*a*/MAT*a or *MATα/MATα* homozygous diploids [2]. Similarly, fission yeast *Schizosaccharomyces pombe* responds to nutritional and cellular stress to initiate conjugation and meiosis (reviewed in [3]).

In *S. cerevisiae*, two key loci control the event of sporulation: *MAT*, the mating-type locus, and *RME*, the regulator of meiosis [4]. Rme1 is a repressor of sporulation in *MAT* a and *MATα* haploids and has multiple targets [5]. In *MAT*a*/MATα* heterozygous diploids, transcription of the *RME1* gene is itself repressed by the products of *MAT*a and *MATα*, which form an a1-*α*2 repressor complex [6]. Rme1 was initially found to regulate the expression of meiotic genes *IME1* and *IME2* (IME for *inducer of meiosis genes)* [4,7,8]. Later, several targets of Rme1 and regulators of the IME genes essential for meiosis progression were identified [9,10,11,12]. Though the *RME1-IME1* control of entry into meiosis was established and well-studied, the exact mechanism of repression by Rme1 was not clear for decades. The discovery of lncRNAs originating from the *IME1* promoter led to the unraveling of a complex regulatory mechanism via non-coding RNA in the context of meiosis and is an important area of current research to understand the complexity of gene regulation in eukaryotes, especially their role in cell fate determination.

## 2. Non-Coding RNA

The eukaryotic genome encodes genetic information in the form of protein-coding genes essential for the survival of an organism and normal functioning. The RNA molecule plays a pivotal role in carrying information from DNA to protein synthesis in the central dogma of molecular biology. Early research limited the potential of RNA as only an information carrier and translator via mRNA, rRNA or tRNA, while DNA remained the focus of molecular biologists. Later, several RNAs that do not essentially code for any functional protein but possessing important regulatory functions came to the fore. It was found that a large percentage of the eukaryotic genome, previously addressed as ‘junk’, ‘dark matter’ or ‘transcriptional noise’, and transcriptionally inactive and non-coding, had important biological functions.

Now it is well established that, apart from the protein-coding genes, transcription also takes place in the intergenic regions, giving rise to several types of non-coding transcripts [13,14]. In addition, the non-coding transcriptome encompasses antisense RNAs and also those associated with coding regions. Non-coding RNAs are broadly classified into housekeeping and regulatory ncRNAs. While the housekeeping ncRNAs comprise abundantly and ubiquitously expressed transcripts with primary cellular functions, such as rRNA, tRNA, snRNA, snoRNA, TERC, trF and tiRNA; the other category encompasses regulatory ncRNAs, namely miRNA, piRNA, siRNA, eRNA and lncRNA, which regulate epigenetic, transcriptional and post-transcriptional gene expression [15]. Another classification is based on the length of the transcripts. Small ncRNAs have sizes less than 200 nt and most often are associated with the 3’ and 5’ regions of protein-coding genes, while long ncRNAs (lncRNA) have sizes above 200 nucleotides and are transcribed often by RNA polymerase II [16]. Some ncRNAs arise from regions that are close to protein-coding genes, such as promoters and the 3′ ends of genes [17,18], while some lncRNA are transcribed antisense to the protein-coding genes [19].

Loss-of-function mutations in the components of the RNA quality control machineries revealed the presence of several distinct classes of non-canonical non-coding RNA in *S. cerevisiae.* In general, mRNA quality control is regulated by the components of exosomes, usually associated with transcription-independent polyadenylation by the TRAMP complex. Cryptic Unstable Transcripts (CUTs) and Stable Unannotated Transcripts (SUTs) were detected/identified/discovered when exoribonuclease Rrp6, the catalytic component of the exosome complex, was deleted [17,20]; SUTs obtained their annotation as ‘stable’ because they are detected even in wild-type cells with a functional nuclear exosome [21]. Xrn1 mutants (*xrn1Δ*) lacking the cytoplasmic 5′-3′ RNA exonuclease revealed another class of non-coding RNA, the Xrn1-sensitive Unstable Transcripts (XUTs) [22]. Similarly, NUTs, Nrd1-unterminated transcripts, are detected upon deletion of Nrd1, a component of the nuclear Nrd1-Nab3-Sen1 termination complex [23]. Several other classifications are in place to categorize non-coding RNA based on the region in the genome from which they are transcribed, their mechanism of action and their functions, etc. [24,25,26]. While these studies underscore the importance of RNA homeostasis pathways in regulating the non-coding transcriptome, the classification based on the RNA processing pathways is somewhat arbitrary as many lncRNAs are targeted by multiple pathways to various extents.

Non-coding RNA are found across species, including viruses, prokaryotes, yeast, plants and animals [27,28,29,30,31]. Different classes of lncRNA species, viz., CUT, XUT and SUTs, have been identified in diverse fungi, including *S. pombe* and *Naumovozyma castellii* [32,33]. The long non-coding RNAs are evolving as interesting subjects of study as they play important biological roles in several organisms. They regulate chromatin organization, modification and remodeling [34]. Importantly, lncRNAs have been shown to be differentially expressed during various developmental stages in several organisms and appear to have critical roles in cellular differentiation and developmental processes [35,36,37].

Precise strand-specific sequencing studies have revealed the abundance of pervasive/antisense transcription in both budding and fission yeasts [17,20,38,39,40,41]. Several studies report that a large number of antisense transcripts arising from the coding regions are associated with function in meiosis [33,38,39,42,43]. As untimely meiosis leads to serious consequences, it is highly regulated in eukaryotes. While the traditional regulators, viz., transcriptional activators and repressors of meiosis, are known and well-studied, the regulation via the RNA stability and non-coding transcriptome is now coming to light, and the functional consequences and the mechanistic basis of regulation remain to be established in many cases. In this article, we review the progress made in understanding the role of the ncRNA-mediated and RNA-homeostasis-based regulation of meiosis in *S. cerevisiae*.

## 3. Meiotic Non-Coding RNA

Evidence of non-coding RNA induced specifically during meiotic development in *Saccharomyces cerevisiae* was first reported around 10 years ago [44]. It was identified that a unique set of ncRNAs, named Meiotic Unannotated Transcripts (MUTs), are transcribed only in a/α diploids undergoing meiosis. During mitotic division, meiotic genes are targeted and degraded by exosome components in both budding yeast and fission yeast [45,46]. Rrp6, the catalytic component of the nuclear exosome, is involved in the degradation of several meiosis-specific transcripts expressed during vegetative growth. The decrease in Rrp6 protein levels within 4–6 h of inducing sporulation leads to the accumulation of several meiosis-specific non-coding RNAs [44]. Advances in technology and the use of high-resolution tilling arrays made it possible to obtain a complete meiotic transcript landscape and expression profiles of budding yeast [44,47,48,49]. The coding and non-coding transcriptome appears highly dynamic in terms of architecture as cells undergo meiosis. While several meiotically induced transcripts were reported, only some of them, such as *IRT1 (SUT 643), IRT2 (MUT1573), IME4-AS*, etc., are currently associated with a role in regulating meiosis. The function of a large fraction of unannotated transcripts remains to be established.

Ime1 is the master regulator of meiosis and activates several early meiotic genes. Early studies have established Rme1 as the main transcriptional repressor of *IME1* that keeps Ime1 levels low in mitotically dividing cells. *IME4*, a regulator of *IME1* expression, is required for diploids to undergo sporulation [12]. *IME4* codes for an mRNA methyl transferase that mediates methylation at N6-adenosine in several RNA species. This post-transcriptional modification is required for the downregulation of *RME1* mRNA and subsequent expression of *IME1* in diploids for undergoing meiosis [11]. The fine-tuning of this transcriptional repression and activation is now shown to be carried out via several additional RNA-based mechanisms. One key regulatory mechanism appears to be the inhibition of meiotic gene transcription via the production of non-coding antisense RNA. *IME4* is an antagonist in activity to *RME1* [12]. *IME4* is regulated by cell-type-specific antisense transcription [50]. While the *MATa/MATα* heterozygous diploids transcribe the sense strand, haploids produce transcripts from the antisense strand, *IME4-AS*, also known as *RME2*. The ‘default’ expression of this transcript in haploids is a result of a strong promoter. In diploids, the a1-*α*2 complex originating from *MAT* loci represses the antisense strand, allowing the transcription of the sense transcript and thus *IME4* expression, leading to transcriptional activation of *IME1* (Figure 1). Another meiosis-specific gene, *ZIP2*, is suppressed by its antisense transcript, *RME3*, during vegetative growth. Similarly, several other meiosis-related transcripts are repressed by antisense transcription during mitotic growth [50,51,52].

Another mechanism of regulation is via the transcription of lncRNA from the promoter regions of meiotic genes. van Werven et al. (2012) reported the role of lncRNA *IRT1* in repressing the expression of *IME1* [53]. In haploids, *IME1* expression is repressed due to the transcription of *IRT1* in cis at the promoter. The binding of Rme1 to its binding sites upstream of the *IRT1* Transcription Start Site (TSS) initiates transcription of the ncRNA, which prevents the binding of transcriptional activators—for example, Pog1—to the *IME1* promoter. While RNA polymerase is transcribing IRT1, Set1 and Set2 histone methyltransferases are recruited, which deposit methylation marks at H3K4 and H3K36, respectively. Set1-mediated dimethylation of H3K4 recruits the Set3C histone deacetylase complex, which contains two histone deacetylases, Hos2 and Hst1 [54]. Similarly, methylation of H3K36 by Set2 recruits the Rpd3C(S) histone deacetylase complex [55]. These histone modifications create a repressive chromatin state at the *IME1* promoter, thereby repressing the expression. However, it appears that several strain backgrounds show some level of Rme1 expression despite the presence of the a1-*α*2 repressor complex in the heterozygous diploid [56,57]. This will eventually cause the transcription of *IRT1* in the diploids. To overcome this transcription, another lncRNA, termed *IRT2*, is transcribed from regions upstream of the *IME1* promoter [58]. In diploids, accumulation of *IME1* induces the transcription of *IRT2*, which reduces the binding of Rme1 to its binding sites upstream of *IRT1* and therefore a reduction in *IRT1* transcript levels is seen (Figure 2). This provides evidence of a feed-forward cascade mechanism where *IME1* induces its own expression by repressing the transcription of *IRT1* and activating the transcription of *IRT2* when cells are ready to undergo meiosis [59].

Interestingly, *IRT2* is also required for the transcription of *IRT1* in haploid *MATa* and *MATα* cells. While *IRT2* transcription in diploids results in compact nucleosome assembly at Rme1 binding sites and hence reduced transcription of *IRT1*, in haploid cell types, basal-level *IRT2* transcription activates *IRT1* transcription. It was recently shown that *IRT2* transcription recruits Rtt109 histone acetyl transferase, which incorporates a H3K56 acetylation mark on the newly synthesized histones at the *IRT2* locus [58]. The H3K56ac mark facilitates nucleosome unwrapping, thereby allowing the binding of transcriptional activators [60]—in this case, Rme1—which in turn activates *IRT1* transcription.

Several studies report the importance of the non-coding transcriptome for a successful meiosis to occur. However, their functions and the molecular mechanisms by which they act are largely unexplored. For example, *SUT367* was identified in a screen for finding functions of non-coding RNA from a deletion library and it was found that deletion of the essential ncRNA *SUT367* leads to overexpression of the adjacent gene *RPL3*, encoding a ribosomal protein, and this prevents spores from germinating post-meiosis [61]. However, the mechanistic basis of this inhibition is not understood.

## 4. RNA Processing Factors Regulate Meiosis

All these mechanisms indicate the importance of the non-coding transcriptome during essential biological processes such as meiosis. The fine-tuning of various players involved in meiosis is a pre-requisite for successful meiosis to occur. The regulation and mechanisms involved in the turnover of meiosis-specific coding and non-coding transcripts are relatively well understood in fission yeast, *Schizosaccharomyces pombe* [62,63,64,65], but remain to be studied in *S. cerevisiae.* In both budding and fission yeasts, a subset of meiosis-specific genes are constitutively expressed during mitotic growth. However, these genes, in the case of fission yeast, are largely under the control of an RNA-binding protein, Mmi1, which binds a conserved region, the Determinant of Selective Removal sequence (DSR), in these transcripts [45,52,66,67]. As soon as the DSR region is transcribed by RNA Pol II, binding of Mmi1 interferes with the 3′ end processing, causing aberrantly cleaved transcripts, resulting in hyperadenylation by Pla1, a polyA polymerase [68]. This hyperadenylation event targets these transcripts to Rrp6 exosome degradation during mitosis [62,67]. Upon initiation of the meiotic program, the inactivation of Mmi1 for inhibiting meiotic mRNA degradation is mediated by *meiRNA,* a lncRNA from the *sme2+* locus. Mmi1, along with an RNA-binding protein, Mei2, and lncRNA *meiRNA* forms a nuclear dot referred to as the Mei2 dot at the *sme2+* locus [45]. Another lncRNA, *mamRNA,* regulates the mutual control of Mei2 and Mmi1 for efficient switching to meiosis. During mitosis, *mamRNA* binding to Mmi1 enables Mmi1 to target Mei2 for degradation via Ccr4-Not-mediated ubiquitination. Conversely, during meiosis, accumulating Mei2 associates with *mamRNA* and inactivates Mmi1 to inhibit the degradation of meiotic mRNA, thus positively regulating meiosis [62,69]. Similarly, meiosis-specific lncRNAs are also regulated by Mmi1-mediated RNAi silencing [68,70]. In *S. pombe, mei4,* encoding a master regulator of meiosis and sexual differentiation, is silenced during vegetative growth by recruitment of the RNA interference complex, the RNA-Induced Transcriptional Silencing complex (RITS). Interestingly, the recruitment of the RITS complex is via the RNA-binding protein Mmi1 associated with the DSR region of the meiotic transcripts [68]. Rrp6, Red1 and Pab2 facilitate the association of the RITS complex to these targets. Association of the RITS complex is thought to establish a repressive chromatin state by methylation of H3K9 by Clr4, histone deacetylase Clr3 and Mit1 [71,72]. Upon initiation of meiosis, localization of RITS to *mei4* is lost, leading to its active expression.

In addition, transcription termination and RNA 3′ end processing are also reported to regulate the differential expression of meiotic mRNA during mitosis and meiosis. In *S. pombe,* one of the targets of Mmi1, the lncRNA *nam1*, is not only targeted for degradation by Mmi1, but transcription is also efficiently terminated by it during vegetative growth. This co-transcriptional termination of *nam1* is essential for the expression of a downstream gene, *byr2+,* that plays a key role in the control of sexual differentiation [70]. In the *nam1-1* mutant defective in binding Mmi1, transcription readthrough of *nam1* into the adjacent locus coding for *byr2* suppresses *byr2* expression and leads to defective sporulation [70]. Interestingly, several 3′ RNA processing and transcription termination factors, such as Pla1 and Dhp1 (the Rat1/Xrn2 homolog in fission yeast), interact with Mmi1 [73,74]. Loss-of-function mutations of these factors revert the sporulation defect in *meiRNA-*deleted cells. On the other hand, in budding yeast, mutants of transcription termination factor Rtt103 and Rai1 are also defective in the formation of spores [75,76]. While the molecular basis of this defect is not known, it is interesting that Rtt103 interacts with the phosphorylated Thr4 of CTD and specifically recruits transcriptional termination machinery to non-coding snoRNAs [77]. It leads to an important question of whether transcription termination factors such as Rtt103 and Rai1 regulate meiosis-specific non-coding RNAs. Rhn1, the Rtt103 homolog in *S. pombe*, downregulates the expression of meiosis-specific genes during vegetative growth and mutants also show defective sporulation, again hinting at a potentially conserved function for such RNA processing factors in regulating the meiotic transcriptome [64,78].

Reports suggest that non-coding RNA are usually the targets of the nuclear exosome, Xrn1-dependent cytoplasmic 5′-3′ degradation and non-sense-mediated decay in *S. cerevisiae* [17,20,21,22,79,80]. However, a huge gap exists in understanding how levels of different ncRNA are orchestrated during meiosis in *S. cerevisiae.* It is likely that RNA processing and termination factors might be predominant modes of regulating meiotic non-coding RNA in *S. cerevisiae* in the absence of RNAi and DSR-Mmi1-dependent regulatory processes.

Cell fate determination is a complex process regulated by multiple interconnected pathways. The diverse mechanisms, ranging from regulation by transcription factors at the promoters and chromatin remodeling at the meiotic genes, to regulation by pervasive non-coding RNA and RNA processing, showcase the importance of the stringent control that is required to determine cell fate decisions. Studying the mechanisms involved in cell fate determination will reveal layers of complex regulation in essential biological pathways that are only now being understood by the research community. Single-cell organisms with simple fates and reduced complexity provide ideal models to investigate the various molecular mechanisms that contribute to this process.

## Figures and Tables

**Figure 1 ncrna-08-00077-f001:**
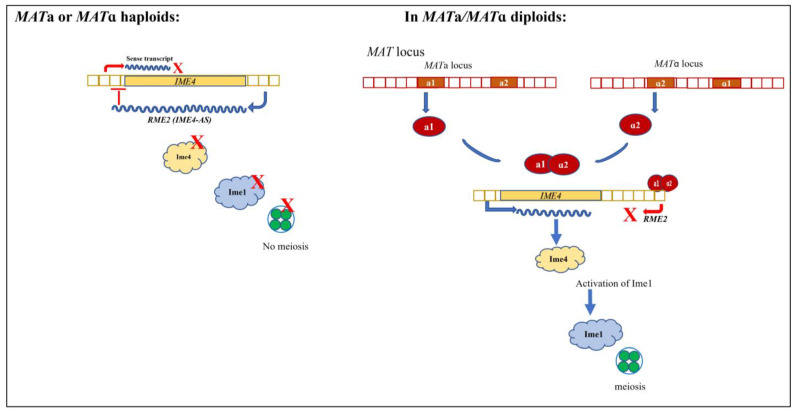
Cell-type-specific control of meiosis via antisense non-coding RNA: Ime4 is an activator of *IME1,* the master regulator of meiosis. In haploid *MAT*a and *MATα* cells, an antisense transcript, *RME2,* is produced from the *IME4* locus. *RME2* represses the expression of Ime4 by potentially preventing the elongation of the *IME4* RNA. Therefore, haploid cells do not progress to meiosis. On the other hand, in heterozygous *MAT*a/*MATα* diploids, a1 and *α*2 expressed from the mating-type *MAT*a and *MATα* locus form the a1-*α*2 repressor complex. This repressor complex binds to binding sites present downstream of the *IME4* locus, thus repressing the antisense transcription. Therefore, in the absence of *RME2,* the *IME4* sense transcript is created and Ime4 is expressed, which can now activate *IME1* expression and initiate meiosis.

**Figure 2 ncrna-08-00077-f002:**
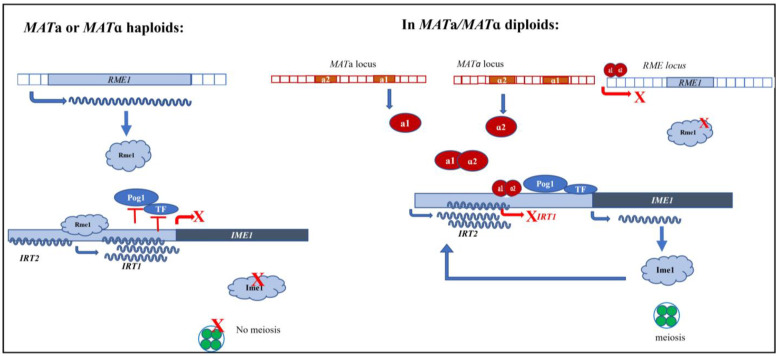
Repression of *IME1* and meiosis by lncRNA *IRT1*: Rme1 negatively regulates the transcription of *IME1,* the master regulator of meiosis. Two long non-coding transcripts, *IRT1* and *IRT2*, arising from the promoter of *IME1*, further modulate the expression of Ime1. In haploid cells, basal-level transcription of *IRT2* enhances the binding of Rme1 upstream of *IRT1* TSS, which activates the transcription of *IRT1*. This transcription inhibits the binding of transcription factors, such as Pog1, and also leads to the deposition of repressive chromatin marks. This suppresses the transcription of *IME1*. Meanwhile, in diploids, the a1-*α*2 repressor complex from *MAT* loci inhibits transcription of Rme1. Transcription of *IRT1* by any leaky Rme1 is prevented by both a1-*α*2 repressor binding to Rme1 binding sites in the *IME1* promoter and also the transcription of *IRT2*. In addition, Ime1 protein also activates its own transcription via *IRT2*. This allows the expression of *IME1* and progression into meiosis.

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
