# Peer review of "RNA-Mediated Regulation of Meiosis in Budding Yeast"

_ncrna, 2022, doi:10.3390/ncrna8060077_

Round 1

Reviewer 1 Report

The authors wrote a short review on the function noncoding RNAs in meiosis in yeasts. They address some of the new recent findings. The review is appropriate in itself but contains errors that I attempted to have highlighted here below. Note that I could not comment on pombe literature as I am not perfectly familiar with that.

“ Expression of Ime4, an essential tran- 123 scriptional activator of IME1, is required for the diploids to undergo sporulation. The fine 124 tuning of this transcriptional repression and activation is now shown to be done via sev- 125 eral RNA-based mechanisms. One key regulatory mechanism appears to be the inhibition 126 of meiotic gene transcription via production of non-coding antisense RNA. IME4 is an 127 antagonist in activity to RME1 [13].”

-        This section contains some factual errors, and is incomplete. Below some suggestions:

Ime4, is RNA methyl transferase (NOT a transcription factor) that methylates mRNAs at n6-methyladosine position. Ime4 may contribute to IME1 transcription by degrading RME1.  https://www.nature.com/articles/s41467-019-11232-7. There is evidence that Ime1 regulates Ime4 https://www.ncbi.nlm.nih.gov/pmc/articles/PMC5100854/

-        General comments. IRT1, IME4-AS, MUT1573 etc……. should be italicized

“While RNA polymerase is transcribing IRT1, it also deposits H3K4 histone methylation 152

via Set3 and Set3C which recruit histone deacetylases Hos2 and Hst1.”

-        Not correct. It should K4 methylation via Set1. The K4 dimethylation marks recruits Set3 complex (SET3C), which contains Set3 and the histone deacetylases Hos2 and Hst1

-        Figure 2. Why is IRT2 not included as activator in the model? I would suggest to include that both repressive and activator roles are important as described in the main text!

“It was recently shown that the reduc- 180

tion in nucleosome occupancy via H3K56 acetylation mark incorporated by Rtt109, in- 181

creased Rme1 binding and this leads to enhanced IRT1 transcription [48].”

Rtt109 is a HAT that directly acetylates K56 of newly synthesized histones. The model is that IRT2 transcription facilitates K56ac incorporation in nucleosomes at the IRT2 locus, which in turn allows recruitment of Rme1.

Perhaps mention that, nuclear exosome also controls budding yeast meiosis.

https://www.ncbi.nlm.nih.gov/pmc/articles/PMC3024698/

https://journals.plos.org/plosone/article?id=10.1371/journal.pone.0107648

Author Response

Response to reviewers

We thank all the reviewers for carefully reading the manuscript and the suggestions made for improving the accuracy and quality of the manuscript. Please find below our response to all the comments.

Reviewer 1:

Comment 1: “Expression of Ime4, an essential transcriptional activator of IME1, is required for the diploids to undergo sporulation. The fine tuning of this transcriptional repression and activation is now shown to be done via several RNA-based mechanisms. One key regulatory mechanism appears to be the inhibition of meiotic gene transcription via production of non-coding antisense RNA. IME4 is an antagonist in activity to RME1 [13].”

This section contains some factual errors, and is incomplete. Below some suggestions:

Ime4, is RNA methyl transferase (NOT a transcription factor) that methylates mRNAs at n6-methyladosine position. Ime4 may contribute to IME1 transcription by degrading RME1.  https://www.nature.com/articles/s41467-019-11232-7. There is evidence that Ime1 regulates Ime4 https://www.ncbi.nlm.nih.gov/pmc/articles/PMC5100854/

Response: We thank the reviewer for pointing out the errors. We have corrected the sentence and have described the regulation more accurately in the revised manuscript.

Comment 2:    General comments. IRT1IME4-AS, MUT1573 etc……. should be italicized

Response: We have italicized all the non-coding RNAs in the revised manuscript.

Comment 3: While RNA polymerase is transcribing IRT1, it also deposits H3K4 histone methylation 152 via Set3 and Set3C which recruit histone deacetylases Hos2 and Hst1.”

-        Not correct. It should K4 methylation via Set1. The K4 dimethylation marks recruits Set3 complex (SET3C), which contains Set3 and the histone deacetylases Hos2 and Hst1

Response: We have expanded the sentences and corrected the error in the revised manuscript.

Comment 4:     Figure 2. Why is IRT2 not included as activator in the model? I would suggest to include that both repressive and activator roles are important as described in the main text!

 Response: We have modified the image to depict the role of IRT2 in haploids

Comment 5: “It was recently shown that the reduction in nucleosome occupancy via H3K56 acetylation mark incorporated by Rtt109, increased Rme1 binding and this leads to enhanced IRT1 transcription [48].”

Rtt109 is a HAT that directly acetylates K56 of newly synthesized histones. The model is that IRT2 transcription facilitates K56ac incorporation in nucleosomes at the IRT2 locus, which in turn allows recruitment of Rme1.

Response: We have rephrased the sentence to accurately describe the recruitment of Rme1.

Comment 6: Perhaps mention that, nuclear exosome also controls budding yeast meiosis.

https://www.ncbi.nlm.nih.gov/pmc/articles/PMC3024698/

https://journals.plos.org/plosone/article?id=10.1371/journal.pone.0107648

Response : It was mentioned  (reference number 46).

Reviewer 2 Report

This is an interesting review focusing on non-coding RNA-mediated regulation of meiosis initiation in budding yeast S. cerevisiae. The authors summarize complicated regulation of expression of meiosis inducer genes, IME1 and IME4, in which non-coding RNAs play crucial roles. 

Overall, the manuscript is appropriate and fairly comprehensive. I have only a few minor comments.

Figure 1. In haploid cells, the RME2 transcript itself seems to repress IME4 transcription. If it has not been demonstrated, the figure needs to be corrected.

Why is the IME4 gene written as IME-4 with a hyphen?

Fission yeast gene names are expressed as an italicized lower-case letter according to general fission yeast nomenclature rules.

lines 206 and 214. MEI4 should be mei4.

line 220. BYR2 should be byr2.

Author Response

Response to reviewers

We thank all the reviewers for carefully reading the manuscript and the suggestions made for improving the accuracy and quality of the manuscript. Please find below our response to all the comments.

Reviewer 2:

Comment 1: Figure 1. In haploid cells, the RME2 transcript itself seems to repress IME4 transcription. If it has not been demonstrated, the figure needs to be corrected.

Response: The figure and legend have been modified to accurately represent the mechanism of Rme2 mediated repression of Ime4

Why is the IME4 gene written as IME-4 with a hyphen?

Response:  The gene notation has been corrected in the revised manuscript.

Comment 2: Fission yeast gene names are expressed as an italicized lower-case letter according to general fission yeast nomenclature rules.

lines 206 and 214. MEI4 should be mei4.

line 220. BYR2 should be byr2.

Response: We have changed the gene names to lower case in the revised manuscript

Reviewer 3 Report

This manuscript reviews the current knowledge on lncRNA-mediated regulation of meiosis in budding yeast. I think it is complementary to the review published last year in Non-coding RNA by Andric & Rougemaille: “Long Non-Coding RNAs in the Control of Gametogenesis: Lessons from Fission Yeast”.

Globally, the manuscript is well written and provides a clear synthesis of the literature.

I only have minor comments, mainly consisting in addition of references.

* Lines 57-58: add references illustrating the pervasive transcription of the genome, at least in yeast (PMID: 21771634, 25956976).

* Lines 69-70: this sentence is misleading and should be rephrased. Many ncRNAs are produced from regions that are ‘close’ from protein-coding genes, such as promoters (many CUTs are produced from bidirectional promoters). Many lncRNAs (such as XUTs) are antisense to protein-coding genes.

* Line 77: the difference between CUTs and SUTs should be explained more carefully. I would not mention their half-life, as this was not experimentally tested. In terms of definition, SUTs were defined as ‘stable’ transcripts as they could be detected in wild-type cells and were not sensitive to the Exosome. However, it was then showed that many of them overlap XUTs (same RNA that would be detectable in wild-type cells though targeted by Xrn1, or distinct RNA isoforms as shown for the lncRNAs antisense to ARG1 – see PMID: 26805575). Ref. 67 also nicely showed that extended isoforms of SUTs can then be targeted by Xrn1.

* Line 89: the sentence is misleading and should be rephrased. If this is true that lncRNAs do not display sequence conservation between species, the different classes are conserved between yeast species. In fact, CUTs and XUTs have been identified in fission yeast (PMID: 29114019, 29914874). CUTs, SUTs and XUTs were also identified in the budding yeast N. castellii (PMID: 31462400). This could be mentioned to provide complete and accurate information to the reader.

* Lines 95-96: more recent references should be added to illustrate the detection of antisense transcription in budding yeast (PMID: 21248844) and fission yeast (PMID: 26883383, 29114019).

* Line 96-98: other studies have reported antisense RNAs associated to meiosis (PMID: 22186733, 29114019, 29914874).

* Lines 204-205: please provide further information about meiRNA and mamRNA. How does mamRNA regulate Mei2 and meiRNA?

* Lines 235-236: there is no evidence for a role of cytoplasmic 3’-5’ degradation in targeting lncRNAs in S. cerevisiae. LncRNAs are targeted by

- the Exosome-dependent 3’-5’ decay (Xu et al., 2009; Neil et al., 2009 – already cited as ref. 16-17 but should also be mentioned here)

- the Xrn1-dependent cytoplasmic 5’-3’ decay (van Dijk et al., 2011 – already cited as ref. 18 but should also be mentioned here)

- the Nonsense-Mediated mRNA decay pathway (PMID: 26805575, 25905671). These references should be cited here.

* Throughout the manuscript, there are some typos that should be corrected (for example, line 130 ‘anti-sense’, line 34 ‘Rem 1’, line 37 double space between ‘the’ and ‘expression’). Please also check that gene and protein names are correctly spelled.

Author Response

Response to reviewers

We thank all the reviewers for carefully reading the manuscript and the suggestions made for improving the accuracy and quality of the manuscript. Please find below our response to all the comments.

Reviewer 3:

Comment 1: Lines 57-58: add references illustrating the pervasive transcription of the genome, at least in yeast (PMID: 21771634, 25956976).

Response: We have included the references and edited text accordingly.

Comment 2: Lines 69-70: this sentence is misleading and should be rephrased. Many ncRNAs are produced from regions that are ‘close’ from protein-coding genes, such as promoters (many CUTs are produced from bidirectional promoters). Many lncRNAs (such as XUTs) are antisense to protein-coding genes.

Response: The sentence has been rephrased in the revised manuscript. And additional references have been included. 

Comment 3: Line 77: the difference between CUTs and SUTs should be explained more carefully. I would not mention their half-life, as this was not experimentally tested. In terms of definition, SUTs were defined as ‘stable’ transcripts as they could be detected in wild-type cells and were not sensitive to the Exosome. However, it was then showed that many of them overlap XUTs (same RNA that would be detectable in wild-type cells though targeted by Xrn1, or distinct RNA isoforms as shown for the lncRNAs antisense to ARG1 – see PMID: 26805575). Ref. 67 also nicely showed that extended isoforms of SUTs can then be targeted by Xrn1.

Response: We have removed half-life from the description and reworded the sentences.

Comment 4:  Line 89: the sentence is misleading and should be rephrased. If this is true that lncRNAs do not display sequence conservation between species, the different classes are conserved between yeast species. In fact, CUTs and XUTs have been identified in fission yeast (PMID: 29114019, 29914874). CUTs, SUTs and XUTs were also identified in the budding yeast N. castellii (PMID: 31462400). This could be mentioned to provide complete and accurate information to the reader.

Response: The sentence has been rephrased and the suggested references have been added.

Comment 5: Lines 95-96: more recent references should be added to illustrate the detection of antisense transcription in budding yeast (PMID: 21248844) and fission yeast (PMID: 26883383, 29114019).

Response: These references have been included in the revised text

Comment 6: Line 96-98: other studies have reported antisense RNAs associated to meiosis (PMID: 22186733, 29114019, 29914874)

Response: The suggested references have been included.

Comment 7: Lines 204-205: please provide further information about meiRNA and mamRNA. How does mamRNA regulate Mei2 and meiRNA?

Response: We have briefly described the mechanism and additional references have been included in the revised manuscript.

Comment 8: Lines 235-236: there is no evidence for a role of cytoplasmic 3’-5’ degradation in targeting lncRNAs in S. cerevisiae. LncRNAs are targeted by

- the Exosome-dependent 3’-5’ decay (Xu et al., 2009; Neil et al., 2009 – already cited as ref. 16-17 but should also be mentioned here)

- the Xrn1-dependent cytoplasmic 5’-3’ decay (van Dijk et al., 2011 – already cited as ref. 18 but should also be mentioned here)

- the Nonsense-Mediated mRNA decay pathway (PMID: 26805575, 25905671). These references should be cited here.

Response: We have corrected the error and in the revised manuscript all references are now cited as suggested.

Comment 9: Throughout the manuscript, there are some typos that should be corrected (for example, line 130 ‘anti-sense’, line 34 ‘Rem 1’, line 37 double space between ‘the’ and ‘expression’). Please also check that gene and protein names are correctly spelled.

Response: We have re-read the manuscript and multiple such errors, including spellings, notations and grammar have been corrected.

Round 2

Reviewer 1 Report

Regarding the following:

"Expres- 134 sion of IME4, an essential regulator of IME1 transcription is required for diploids to un- 135 dergo sporulation [13]. IME4 codes for an mRNA methyl transferase that mediates meth- 136 ylation at N6- adenosine of several RNA species. This post-transcriptional modification 137 is required for downregulation of RME1 mRNA and subsequent expression of IME1 in 138 diploids for undergoing meiosis[12]."

""Expres- 134 sion of IME4, an essential regulator of IME1 transcription is required for diploids to un- 135 dergo sporulation [13]." 

This sentence is not accurate in my view. ime4deletion affects IME1 expression. This observation does not say anything about transcription and whether it is a direct effect. As far is understood Ime4 does not regulate transcription, at least there is no evidence for that. The authors can be a bit more concise here. 

 I have no additional comments. 

Author Response

Cooment:

"Expression of IME4, an essential regulator of IME1 transcription is required for diploids to undergo sporulation [13]." 

This sentence is not accurate in my view. ime4deletion affects IME1 expression. This observation does not say anything about transcription and whether it is a direct effect. As far is understood Ime4 does not regulate transcription, at least there is no evidence for that. The authors can be a bit more concise here. 

Response:

We have modified the sentence to say "expression" rather than "transcription".

It now reads:

IME4, a regulator of IME1 expression, is required for diploids to undergo sporulation(12). IME4 codes for an mRNA methyl transferase that mediates methylation at N6- adenosine of several RNA species. This post-transcriptional modification is required for the downregulation of RME1 mRNA and subsequent expression of IME1 in diploids for undergoing meiosis.